# Strain fingerprinting of exciton valley character in 2D semiconductors

Abhijeet M. Kumar [1,7], Denis Yagodkin [1,7], Roberto Rosati [2], Douglas J. Bock[1], Christoph Schattauer[3], Sarah Tobisch [3], Joakim Hagel[4], Bianca Höfer[1], Jan N. Kirchhof [1,5], Pablo Hernández López [6], Kenneth Burfeindt[1], Sebastian Heeg [6], Cornelius Gahl[1], Florian Libisch [3], Ermin Malic [2] & Kirill I. Bolotin [1] ✉

Intervalley excitons with electron and hole wavefunctions residing in different valleys determine the long-range transport and dynamics observed in many semiconductors. However, these excitons with vanishing oscillator strength do not directly couple to light and, hence, remain largely unstudied. Here, we develop a simple nanomechanical technique to control the energy hierarchy of valleys via their contrasting response to mechanical strain. We use our technique to discover previously inaccessible intervalley excitons associated with K, Γ, or Q valleys in prototypical 2D semiconductors $WSe_2$ and $WS_2$. We also demonstrate a new brightening mechanism, rendering an otherwise "dark" intervalley exciton visible via strain-controlled hybridization with an intravalley exciton. Moreover, we classify various localized excitons from their distinct strain response and achieve large tuning of their energy. Overall, our valley engineering approach establishes a new way to identify intervalley excitons and control their interactions in a diverse class of 2D systems.

Two-dimensional semiconductors from the family of transition metal dichalcogenides (TMDs) host a wide variety of excitons, Coulomb-bound electron-hole pairs. These excitons $X_{AB}$ with hole and electron wavefunctions residing in valleys K, K', Γ (A) and K, K', Q (B), respectively, determine the optical response of TMDs[1,2]. Much of the early research was devoted to intravalley excitons from K valley ($X_{KK}$) that directly couple to light, exhibiting prominent features in the optical absorption and emission spectra[1,3–5]. However, these "bright" species represent only a small subset of excitons in TMDs. Recent studies show that a much more diverse class of momentum-indirect intervalley excitons have an outsized effect on the optoelectronic properties of TMDs. These excitons constitute the lowest excited states of several TMDs, weakly interact with light, and show orders of magnitude long lifetimes compared to their direct counterparts, rendering them prime

candidates for the realization of exciton condensate[2,6–12]. Their interactions and interconversion with the intravalley excitons drive the temporal dynamics and long-range transport observed in these materials[6,13]. Moreover, the intervalley excitons also offer new opportunities for spin/valley-tronics owing to the coupled spin and valley degrees of freedom[14].

Despite multiple studies suggesting the defining role of intervalley excitons in the optical and transport properties of TMDs, they are much less studied compared to their intravalley counterparts. The key challenge lies in the conventional optical spectroscopy approaches, which are hampered by the lack of momentum resolution. Consequently, they cannot differentiate between the excitons residing in different valleys, exacerbated by the low oscillator strength of these quasiparticles[15]. Additionally, some intervalley excitons such as $X_{KK'}$ or

[1]Department of Physics, Freie Universität Berlin, Arnimallee 14, Berlin, Germany. [2]Philipps-Universität Marburg, Mainzer Gasse 33, Marburg, Germany. [3]Institute for Theoretical Physics, TU Wien, Wiedner Hauptstraße 8-10, Vienna, Austria. [4]Department of Physics, Chalmers University of Technology, 412 96 Gothenburg, Gothenburg, Sweden. [5]Kavli Institute of Nanoscience, Delft University of Technology, Lorentzweg 1, 2628 CJ Delft, Delft, The Netherlands. [6]Institute for Physics and IRIS Adlershof, Humboldt-Universität Berlin, Newtonstraße 15, Berlin, Germany. [7]These authors contributed equally: Abhijeet M. Kumar, Denis Yagodkin. ✉e-mail: kirill.bolotin@fu-berlin.de

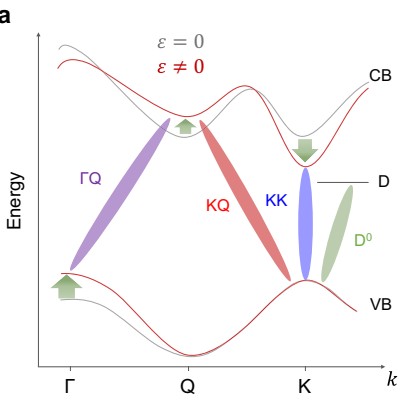

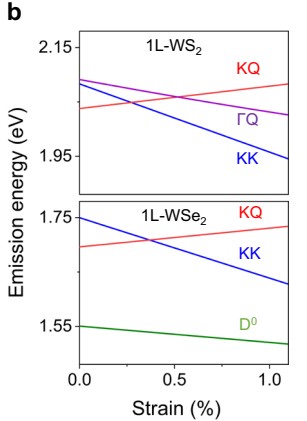

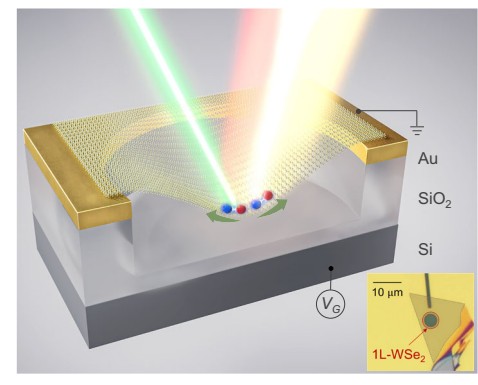

**Fig. 1 | Influence of strain on excitons in TMDs and experimental technique.**
**a** Schematic band structure of 1L-WSe$_2$ under zero (grey) and tensile (red) strain. Different valleys respond to strain differently (green arrows). **b** Calculated energies of various intra- and inter-valley excitons vs. biaxial strain in 1L-WS$_2$ and 1L-WSe$_2$. **c** Straining technique: an applied gate voltage ($V_G$) induces biaxial strain $\varepsilon$ in the center of a suspended TMD monolayer via electrostatic forces. The arrows denote the tensile strain. The inset shows an optical image of a monolayer WSe$_2$ suspended over a circular trench.

X$_{KQ}$ lie in the energetic proximity of a series of tightly-bound KK excitons (trions, biexcitons, etc.), further complicating their study. In principle, techniques such as angle-resolved photoemission spectroscopy (ARPES) or electron energy loss spectroscopy (EELS) provide the momentum resolution needed to identify the intervalley states[16–20]. However, the typical separation between the excitonic species ( ~10 meV) in TMDs is smaller compared to the energy resolution of these techniques. These approaches also require large-area devices and pose challenges in integration with external stimuli such as electric and magnetic fields.

In this work, we apply a nanomechanical approach to tune the energy hierarchy of valleys and identify the valley character of excitons in TMDs. We tackle the challenges outlined above by studying the response of excitonic photoluminescence (PL) peaks to mechanical strain at cryogenic temperatures. We show that an exciton's energy changes with strain at a rate characteristic of the valleys in which its electron and hole wavefunctions reside, allowing straightforward identification of the exciton valley character−"valley fingerprinting". Moreover, a distinct strain response implies that at a specific strain value, the intervalley excitons can be brought into energetic resonance with the bright intravalley excitons. The hybridization between these species brightens the dark states allowing their optical detection. We use our technique to directly observe and identify previously hypothesized, but optically inaccessible intervalley KQ and ΓQ excitons. In addition, we fingerprint the fine structure of well-known KK excitons and several types of defect-related excitons. Finally, we achieve in-situ strain tuning of quantum-confined excitons associated with single-photon emitters by up to 60 meV. In a larger context, our work establishes strain engineering as a universal tool to identify the valley character of excitons, tune their energy hierarchy, and exploit inter-excitonic interactions in 2D materials.

## Results

### Theoretical predictions for excitonic strain response

We start by analyzing the strain response of various intra- and inter-valley excitons in TMDs theoretically. First-principles calculations show[2,21–23] that the conduction band (CB) minima ($E^{CB}$) and the valence band (VB) maxima ($E^{VB}$) in different valleys (e.g., K, Q, Γ) as well as the defect state D react differently to an applied strain, $\varepsilon$ (tensile biaxial strain unless stated otherwise). For example, the CB at the K valley shifts down in energy relative to the K valley VB with increasing tensile strain, the energy of the defect states remains nearly strain-independent, and the CB at the Q valley (also referred to as Λ) shifts up

in energy (Fig. 1a). The behavior of each valley is governed by the distinct strain dependence of the overlap of electronic orbitals constituting its wavefunctions, Supplementary Fig. S1[21]. In general, the energy of an exciton X$_{AB}$ is given by $E_{AB}^X = E_B^{CB} - E_A^{VB} - E_{AB}^{bind}$, with the last term representing the exciton binding energy. Since the binding energy is only weakly strain-dependent, especially in K and Q valleys (Supplementary Fig. S1), the strain response of an exciton is predominantly defined by its constituent valleys[23,24]. The calculated strain response of various excitons in 1L-WSe$_2$ and 1L-WS$_2$ supports this intuition (Fig. 1b; see Supplementary Notes S1 and S2 for calculation details). For example, K/K'-valley excitons in 1L-WSe$_2$ are predicted to redshift with a strain gauge factor $\Omega_{KK} = \Delta E_{KK}^X/\Delta\varepsilon = 111$ meV/%. The ΓQ excitons also redshift in 1L-WS$_2$, although at a lower rate $\Omega_{\Gamma Q} = 60$ meV/%. A ΓQ exciton in 1L-WSe$_2$ is expected to lie ~270 meV above the KK exciton and therefore is not considered in our study, Supplementary Fig. S1[2]. The KQ excitons in 1L-WSe$_2$ shift in energy under tensile strain with $\Omega_{KQ} = -34$ meV/% while a localized defect-related exciton D$^0$ is only weakly strain-dependent, $\Omega_{D0} = 10-30$ meV/%.

The calculations in Fig. 1b reveal the following key traits. First, from the strain response of an exciton alone, in principle, one can determine its valley character. Second, because different intra- and inter-valley excitons shift in energy with strain at different rates, we can control the energy separation and, hence, the coupling between them. This, in turn, enables novel interactions such as brightening of an otherwise dark X$_{KQ}$ or D$^0$ exciton via hybridization with a bright X$_{KK}$ exciton[6,25,26]. While these ideas have been applied in the past to examine the strain response and hybridization between intervalley excitons[6,27–34], most strain-engineering techniques function only at room temperature. Under these conditions, temperature-related exciton linewidth broadening and thermal dissociation severely limit the range of accessible excitonic species. Examining the full range of intervalley excitons and studying their interactions, therefore, requires low-disorder TMD devices with in-situ strain control up to a few percent at cryogenic temperature.

### Experimental realization of exciton valley fingerprinting

To address these requirements, we use the electrostatic straining technique we recently developed[26] (Fig. 1c). A monolayer flake is suspended over a circular trench in a Au/SiO$_2$/Si substrate where an applied gate voltage ($V_G$) induces biaxial strain in the center of the membrane via electrostatic forces. The device is placed inside a cryostat ($T = 10$ K), while its PL response to an optical excitation is

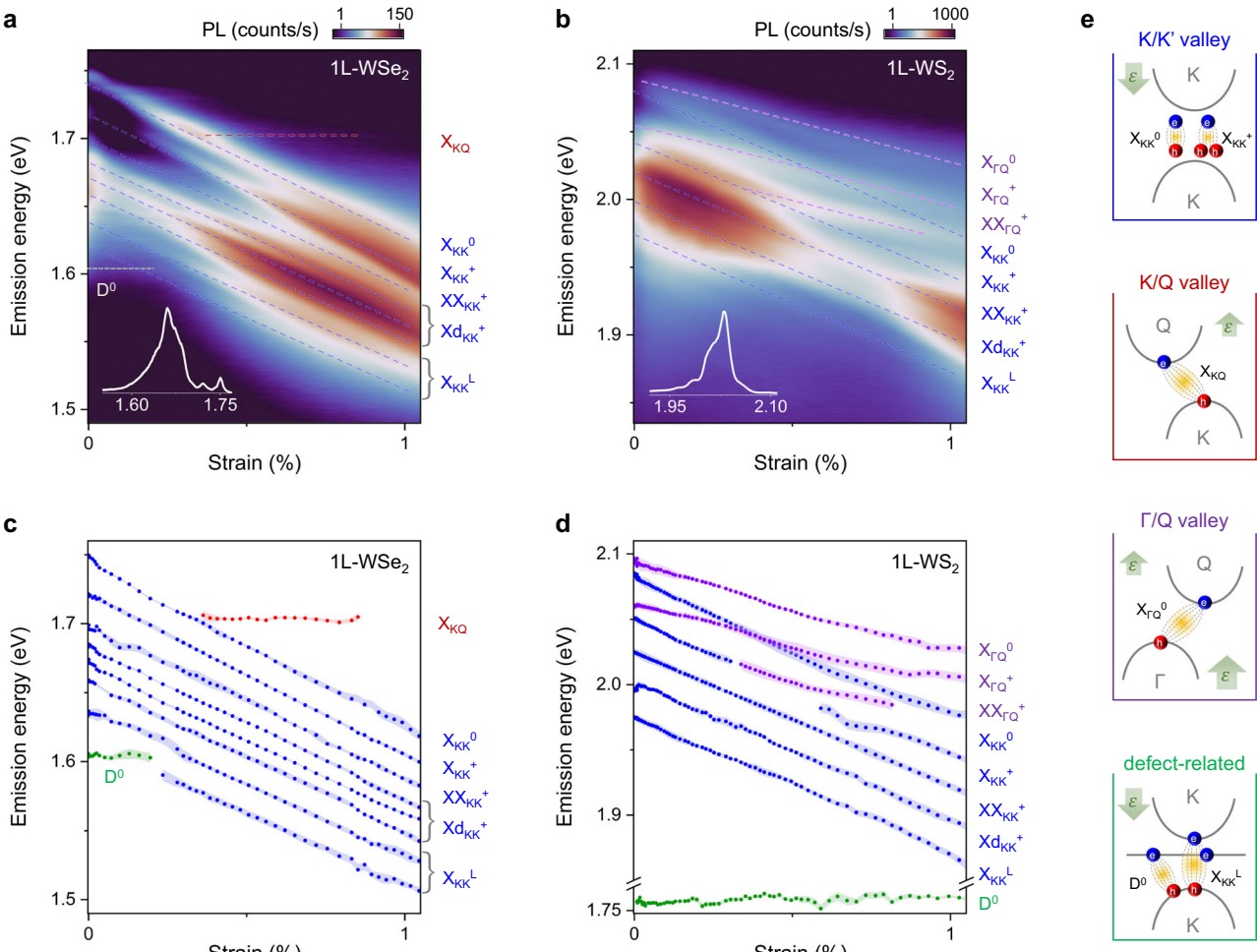

**Fig. 2 | Strain response of excitons and identification of their valley character.** **a**, **b** False color map of PL vs. strain (log-scale) in 1L-WSe$_2$ and 1L-WS$_2$ devices at 10 K. Dashed lines highlight the strain-dependent excitonic peaks: neutral excitons (X$^0$), trions (X$^+$), charged biexcitons (XX$^+$), dark trions and their phonon replicas (Xd$^+$), defect excitons (D$^0$), and excitons bound to defects (X$^L$). Insets show the PL spectra from devices at $\varepsilon = 0$. **c**, **d** Extracted peak positions vs. strain of various excitons from Fig. 2a and b, respectively. The shadows denote uncertainties in the exciton peak positions. The groups of peaks corresponding to a specific strain gauge factor are color-coded. The analysis of the gauge factors allows valley fingerprinting of the corresponding excitons. Note, that a pre-strain may influence the energy positions of the observed excitons[66]. **e** Cartoons depicting valley compositions of KK (blue), KQ (red), ΓQ (purple) and defect-related (green) excitons. Green arrows indicate a strain-induced shift of the K, Q, and Γ valleys with respect to the K valley in the VB.

measured as a function of strain in the center of the suspended flake (see "*Methods*"). The induced strain is tensile and symmetric for both polarity of $V_G$ due to the attractive nature of the electrostatic force (Fig. S2). The applied strain reaches 1.5% at $T = 10$ K, limited by the dielectric breakdown of SiO$_2$.

Figure 2a and b show photoluminescence emission spectra vs. strain of 1L-WSe$_2$ and 1L-WS$_2$, respectively (see Supplementary Note S4, and Supplementary Fig. S3 for strain determination). Both material systems exhibit complex spectra with an abundance of excitonic peaks (insets of Figs. 2a, b). Some well-known excitons such as neutral and charged excitons (X$^0_{KK}$, X$^+_{KK}$), charged biexcitons (XX$^+_{KK}$), dark trions (Xd$^+_{KK}$) and their phonon replicas can be identified at zero strain by comparing their peak positions and power dependence with previous reports[35–37] (Supplementary Figs. S4, S5). Once the strain is applied, different groups of peaks exhibit distinct strain dependence. We extract the energy positions of various excitonic peaks vs. strain and color-code each group of peaks based on their strain dependence (Figs. 2c, d; see Supplementary Note S5 for fitting procedure). We now proceed to assign these peaks to the excitons originating from specific valleys of the electronic band structure.

## KK valley excitons

We first focus on the largest group of peaks (X$^0_{KK}$, X$^+_{KK}$, XX$^+_{KK}$, Xd$^+_{KK}$) shifting down in energy with gauge factor $\Omega_{KK} = 118 \pm 6$ meV/% in 1L-WSe$_2$ and $\Omega_{KK} = 102 \pm 13$ meV/% in 1L-WS$_2$ (blue in Figs. 2c, d, and Supplementary Note S3). This is the shift expected for an optical transition between the VB and CB at the K/K' valley (blue in Fig. 1b), in agreement with previous reports[28,30–32,38]. We find very close gauge factors for species within the KK group (Supplementary Fig. S4). This similarity suggests that i) carrier density-related changes in the exciton emission energy are an order of magnitude smaller compared to strain ($\Delta E(n) < 10$ meV for $n < 1.5 \times 10^{12}$ cm$^{-2}$, see Supplementary Fig. S4d and Supplementary Note S6)[39]; ii) strain-related changes in the phonon energies are minor since the energy spacing between the excitons and their replicas depends on the phonon energy[24,40]; and iii) effective masses near the K valley are nearly strain-independent (as suggested by theory, see Supplementary Fig. S1) since the binding energy of the biexciton (determined from the energy difference between X$_{KK}$ and XX$^-_{KK}$) is affected by strain only via the effective mass. We note that the carrier density changes also influence the intensity and the linewidth of the excitonic features (Supplementary Fig. S4), which is outside the scope of our study.

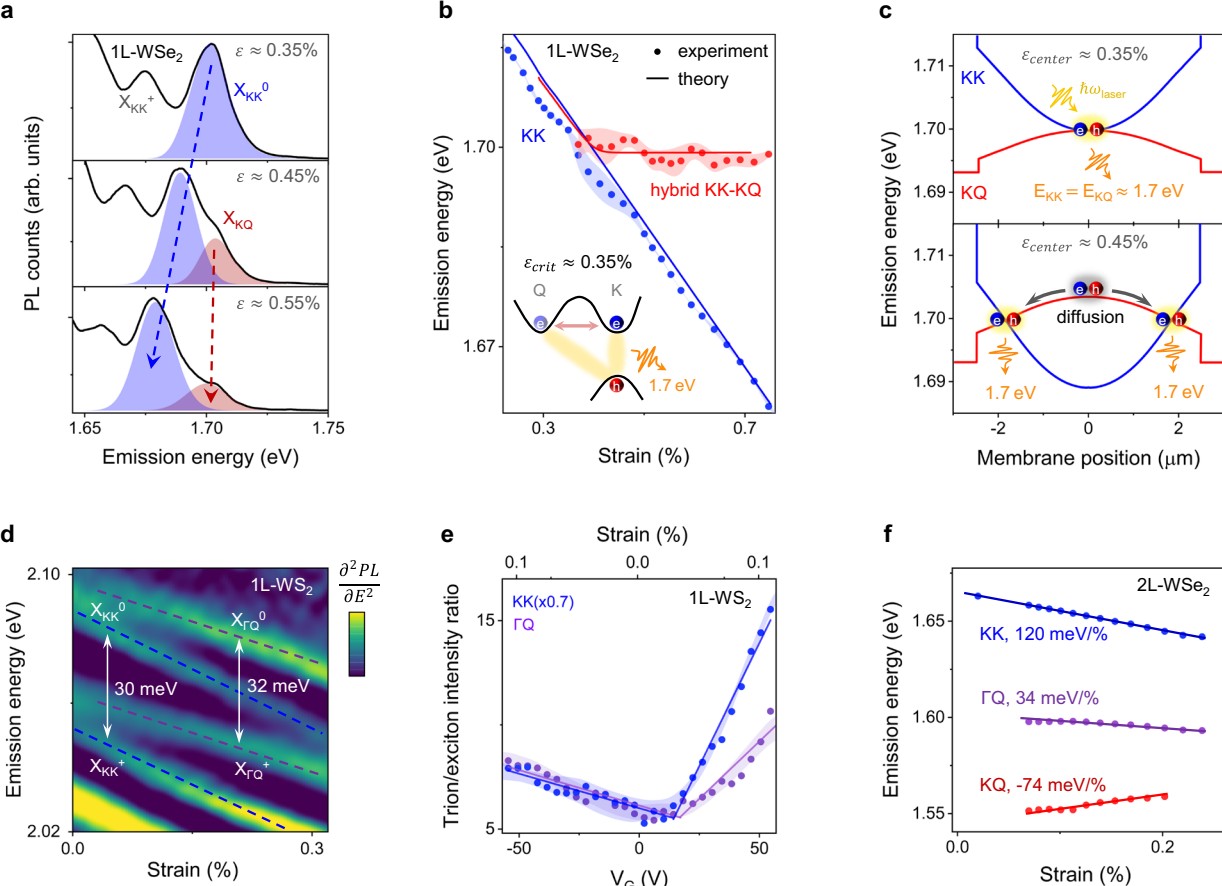

**Fig. 3 | Momentum-indirect Q-valley excitons in WSe$_2$ and WS$_2$. a** PL spectra in a 1L-WSe$_2$ at selected strain values. Above a critical strain value $\varepsilon_{crit} \geq 0.35\%$, a strain-independent feature ($X_{KQ}$, red Gaussian) emerges on the higher energy side of $X_{KK}^0$ (blue Gaussian). **b** Extracted emission energy of $X_{KK}^0$ (blue) and $X_{KQ}$ (red) in 1L-WSe$_2$ vs. strain. The solid lines are the theoretically predicted emission energy of the KK and the hybrid KK-KQ excitons. The inset depicts the hybridization scenario and the corresponding emission. **c** Simulated energies of the KK and KQ excitons vs. position on a line cut across the membrane for two different values of $\varepsilon_{center}$. A strain inhomogeneity, $\varepsilon(x = \pm 1\,\mu m) \approx 0.9 \times \varepsilon_{center}$, causes spatially dependent resonance condition $E_{KK} = E_{KQ}$. A KQ exciton diffuses (anti-funnels) and emits light from

the position within the membrane where the resonance conditions are achieved. **d** False color map of $d^2PL/dE^2$ vs. strain in 1L-WS$_2$ highlighting individual excitonic peaks. Blue and purple lines indicate the KK and $\Gamma Q$ excitons, respectively. The energy difference between a pair of KK peaks and a pair of $\Gamma Q$ peaks corresponds to the binding energy of charged excitons (trions). **e** Ratio of the areas under the trion and exciton peaks vs. $V_G$ for KK (blue) and $\Gamma Q$ (purple) excitons in weak strain regime ($\varepsilon \leq 0.2\%$). For $V_G \geq +20$ V, this ratio for KK and $\Gamma Q$ excitons increases sharply suggesting that the Fermi level has entered the CB. **f** Extracted exciton peak positions vs. strain and corresponding strain gauge factors for KK, $\Gamma Q$, and KQ excitons in a 2L-WSe$_2$.

## KQ excitons

We now turn to the nearly strain-independent peak at 1.70 eV in 1L-WSe$_2$ (red in Fig. 2c). To assign the nature of this feature, we plot PL spectra at selected strain values in a second WSe$_2$ device (Fig. 3a, this sample is different from Fig. 2). While $X_{KK}^0$ (blue shaded) and $X_{KK}^+$ redshift with strain, a new feature (red shaded) appears on the high-energy side of $X_{KK}^0$ above a critical strain value, $\varepsilon_{crit} \approx 0.35\%$. This peak exhibits a linear power dependence suggesting its free excitonic character (Supplementary Fig. S5) and emerges exactly at the strain value at which the $X_{KK}^0$ and $X_{KQ}$ excitons are predicted to come into resonance (0.35% in Fig. 1b; $E_{KK} = E_{KQ} = 1.705$ eV). We therefore suggest that the 1.70 eV peak corresponds to a hybridized state of KQ and KK excitons. In this scenario, a normally dark KQ exciton acquires oscillator strength through resonant hybridization with a bright KK exciton with which it shares the hole wavefunction (inset, Fig. 3b). The emerging hybrid state should emit only in the vicinity of $\varepsilon_{crit}$ while having a vanishing oscillator strength otherwise.

Notably, we observe that the peak at 1.70 eV persists for the applied strain exceeding 0.35% and its energy remains nearly strain-independent afterward (Fig. 3b). We ascribe this behavior to a slight parabolic strain variation in our membrane around the maximum

reached in the center: $\varepsilon(x = \pm 1\,\mu m) \approx 0.9 \times \varepsilon_{center}$ (Fig. 3c, and Supplementary Fig. S6). With increasing $V_G$, the strain first reaches the critical value in the center of the membrane (Fig. 3c, top panel). When the applied strain $\varepsilon_{center}$ exceeds 0.35%, the condition $\varepsilon_{crit} \approx 0.35\%$ required for the KK-KQ hybridization progressively shifts outward from the center of the membrane in a donut-shaped region (Fig. 3c, bottom panel). Therefore, the emission at 1.70 eV persists even for $\varepsilon_{center} > 0.35\%$ and remains strain-independent (Fig. 3a, b). Our theoretical many-particle model for the KK-KQ hybridization accounting for a strain inhomogeneity ($\Delta\varepsilon/\varepsilon_{center}$ approximated by a Gaussian distribution of FWHM 6.1 $\mu m$) and $X_{KQ}$ diffusion[6] confirms the strain-independent behavior (solid lines in Fig. 3b; see Supplementary Note S3 for details). A decrease in the hybrid exciton intensity in the center and a shift of the intensity maximum outward from the center for $\varepsilon_{center} > \varepsilon_{crit}$ further supports our model (Supplementary Fig. S7). In addition to the resonant brightening, a KQ exciton emission may originate via a non-resonant mechanism, e.g. phonon-assisted emission[36] or scattering on a defect, which is not included in our model[34]. However, the distinctive traits of a KQ exciton such as the unique strain response of the emission energy and intensity are accurately captured by our model, and cannot be explained by a non-resonant mechanism

(Supplementary Fig. S7). We also observe signatures of a similar peak in 1L-WS$_2$, however, in that material, an energetic proximity of KQ, ΓQ, and ΓK excitons makes the hybridization mechanism more intricate (Supplementary Fig. S8). We note that the KQ excitons were previously reported in ARPES measurements[16] and invoked to explain exciton transport[6] and dynamics[19,20,36,41] but, to the best of our knowledge, have never been directly seen in a monolayer before via optical spectroscopy techniques.

While the strain response of the KQ exciton in a monolayer is governed by hybridization, its fingerprinting is more straightforward in a bilayer WSe$_2$ (Fig. 3f). Here, the KQ emission is more intense due to an abundance of phonons mitigating excess momentum and a significantly lower energy of $X_{KQ}$ compared to $X_{KK}$ ensuring its higher population[9]. Indeed, we observe a low-intensity peak ~120 meV below $X_{KK}^0$ shifting up in energy as predicted for a KQ exciton[42] (Supplementary Figs. S1, S9).

## ΓQ excitons

According to our calculations, the ΓQ and the KK excitons in unstrained WS$_2$ are nearly resonant, however, can be distinguished under strain due to distinct gauge factors (Fig. 1b). To this end, we note the group of three peaks in Fig. 2d (purple points) red shifting with gauge factor $\Omega_{\Gamma Q} = 63 \pm 8$ meV/%. This gauge factor as well as the energy of the states match theoretical expectations for the ΓQ excitons (Fig. 1b). Within this group, we assign the two highest lying states as neutral ($X_{\Gamma Q}^0$) and charged ΓQ excitons ($X_{\Gamma Q}^+$), respectively. To support our assignment, we first note that the two states are separated by ~32 meV, a value similar to the $X_{KK}^+$ binding energy of 30 meV[43] (Fig. 3d). Second, these features exhibit the characteristic behavior of the charged states, i.e., an increase of neutral to charged exciton conversion with increasing carrier density (Fig. 3e). Finally, we suggest that the lowest-energy ΓQ state ( ~52 meV below $X_{\Gamma Q}^0$ in Fig. 2d) is of biexcitonic nature since it shows a super-linear dependence of PL on the excitation power (PL $\propto P^{1.36}$, Supplementary Fig. S5). Further work is needed to pinpoint the exact configuration and brightening mechanism of this state[44]. We highlight that our suspended devices are ideally suited to study ΓQ excitons: the $X_{\Gamma Q}$ energy in a supported device is affected by screening[45], and is predicted to lie ~50 meV higher with significantly weaker emission, rendering their observation challenging (Supplementary Note S1). To the best of our knowledge, this is the first experimental observation of ΓQ excitons in monolayers.

Bilayer WSe$_2$ is another system where ΓQ excitons have been theoretically predicted, showing near degeneracy between the Γ and K valleys at the VB, $E_{\Gamma Q} - E_{KK} \approx 50$ meV (Supplementary Fig. S1). Indeed, our PL data for bilayer WSe$_2$ indicate a state ~70 meV below $X_{KK}^0$ shifting down in energy with a rate of ~34 ± 8 meV/%, consistent with the expected gauge factors for ΓQ excitons in this material[9] (Supplementary Figs. S1, S9). We note that, unlike in the monolayer case, additional effects such as heterostrain or change in layer separation may complicate the strain response of bilayers.

## Localized excitons

Having identified most of the free excitons, we now proceed to the emission features of localized states distinguished by sublinear power dependence (Supplementary Fig. S5). The energy of the peak labeled $D^0$ in Fig. 2c (green) is nearly strain-independent ($\Omega_{D0} = 8 \pm 10$ meV/%), suggesting that the valleys hosting the corresponding electron/hole wavefunctions shift with strain at nearly equivalent rates. Comparison with Fig. 1b allows us to identify $D^0$ as a chalcogen-vacancy-related defect exciton, consistent with previous study[26]. This state involves an optical transition between the VB at the K-point and the momentum-delocalized defect state below the CB (Fig. 1a). Interestingly, also the peaks ~120 meV and ~150 meV below $X_{KK}^0$, labelled $X_{KK}^L$, show sublinear power dependence. These peaks, however, exhibit a gauge factor of 106 ± 4 meV/%, close to that of KK excitons. We therefore attribute the

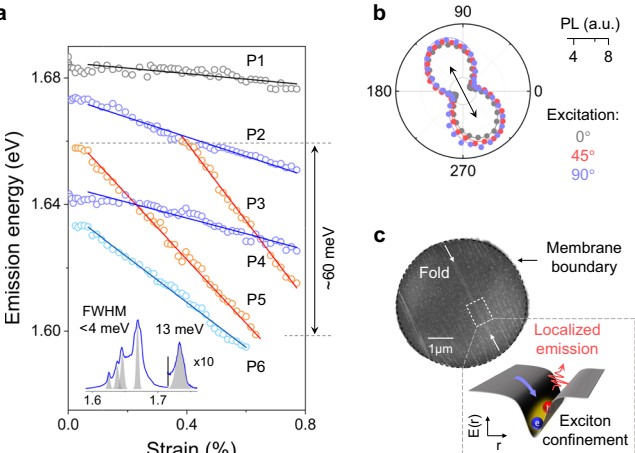

**Fig. 4 | Quantum-confined states in 1L-WSe$_2$. a** Extracted energy positions vs. strain for sharp peaks in 1L-WSe$_2$ (device 3), color-coded by gauge factor. The inset shows a PL line cut from the unstrained device. Note several sharp features with FWHM up to 4 times narrower compared to $X_{KK}^0$ (Supplementary Fig. S10). **b** Polarization-resolved emission of the peak P4 for three different directions of excitation polarization (grey, red, blue). The emission exhibits a preferred direction (black arrow), and is independent of the excitation polarization. **c** Top: SEM image of the membrane reveals several sub-$\mu m$ folds along the emission direction of P4; Bottom: a cartoon depicting localized emission from an exciton trapped in a sharp potential.

peaks, $X_{KK}^L$, to the recombination of a neutral KK exciton bound to a defect. Indeed, states in the same energy range have been recently ascribed to defect-bound excitons[35].

A similar scenario is observed in 1L-WS$_2$ (Fig. 2d): the strain-independent peak $D^0$ appears near 1.76 eV, and the $X_{KK}^L$ peak near 1.97 eV (at $\varepsilon = 0$). The energy difference between $D^0$ and $X_{KK}^0$ is larger in WS$_2$ compared to WSe$_2$ ( ~340 vs. ~150 meV), suggesting different defect energy levels involved in the corresponding optical transitions (Supplementary Fig. S1). We see that, in principle, the strain response allows distinguishing between various types of defect-related excitons including two-particle (e.g., $D^0$) or three-particle ($X_{KK}^L$) states.

In addition to $D^0$ and $X_{KK}^L$ consistently seen in multiple devices, in some devices we observe additional localized exciton peaks. Figure 4 shows strain response in one such WSe$_2$ sample where we identify a group of closely lying peaks labeled P1−P6 with various gauge factors in the range 10−100 meV/%. We note that these peaks are ~4 times narrower than $X_{KK}^L$ (inset in Fig. 4a), have a preferred polarization of emission independent of the excitation (Fig. 4b), and show sublinear power dependence (Supplementary Fig. S10). These features are telltale signs of extensively studied[46–50] single-photon emitters in WSe$_2$. A locally inhomogeneous strain profile has been suggested as the critical requirement for the formation of such emitters via exciton confinement[25,26]. Indeed, the scanning electron microscope (SEM) image of this device shows the presence of folds across the membrane (Fig. 4c, Supplementary Fig. S10) that could lead to sharp (up to 0.02 %/nm) strain gradients[51,52]. Notably, the PL polarization from peaks P1−P6 is oriented along these folds confirming exciton confinement (Fig. 4b). The variation in the gauge factors is then likely associated with the effect of the externally applied strain on the inhomogeneous strain present in the fold. Crucially, the energy of the emitters P1−P6 can be tuned by up to 60 meV under the application of strain. That is the largest reported tuning for such states, to the best of our knowledge[53,54], opening a pathway to engineered quantum emitters. Moreover, two distinct states can be brought into an energetic resonance at specific strain values resulting in an enhanced emission (Supplementary Fig. S10), possibly caused by exciton population redistribution[55]. These observations suggest the potential of confined

**Table 1 | The measured emission energy at zero applied strain ($E(\varepsilon = 0)$), the strain gauge factor ($\Omega$) and the power law exponent ($\alpha$) for several excitons in WSe$_2$ and WS$_2$**

| | WSe$_2$ | | | | | | WS$_2$ | | | | | | | |
|---|---|---|---|---|---|---|---|---|---|---|---|---|---|---|
| | KK | | | | KQ | Defect | KK | | | | ΓQ | | | Defect |
| | $X^O$ | $X^+$ | XX | $X^L$ | $X^O$ | $D^O$ | $X^O$ | $X^+$ | XX | $X^L$ | $X^O$ | $X^+$ | XX | $D^O$ |
| $E(\varepsilon = 0)$, eV | 1.74 | 1.72 | 1.70 | 1.64 | 1.69* | 1.60 | 2.08 | 2.05 | 2.03 | 1.97 | 2.09 | 2.06 | 2.04 | 1.76 |
| $\Omega$, meV/% | 118 | 114 | 113 | 103 | –34* | 8 | 102 | 101 | 100 | 100 | 68 | 55 | 56 | –6 |
| $\alpha$ | 0.97 | 1.11 | 1.42 | 0.70 | 0.95 | 0.66 | 0.97 | 0.99 | 1.33 | 0.86 | 1.01 | 1.18 | 1.36 | 0.50 |

Note, that a pre-strain may influence the $E(\varepsilon = 0)$. Theoretical results are noted with *.

states with engineered strain gauge factors to control exciton hybridization. In the future, $g^2$ measurements will be of particular interest for proving the indistinguishability of such resonant states[56].

## Discussion

To summarize, we established mechanical strain as a powerful tool to brighten the dark intervalley excitons and fingerprint their valley character. In our approach, in-situ control over mechanical strain at low temperatures is the key to unveil the complex excitonic landscape of 1L-WSe$_2$, 1L-WS$_2$, and 2L-WSe$_2$. We identified previously inaccessible ΓQ excitons/trions, KQ excitons in addition to the defect-related excitons and established their strain gauge factor (Table 1). We also identified a brightening mechanism for the normally dark KQ excitons via strain-driven hybridization with bright excitons. We note that our current samples have broader excitonic linewidth compared to the state-of-the-art hBN-encapsulated devices[35,39]. Superior samples with narrower linewidth will enhance the control over closely lying excitonic species. Devices with engineered strain inhomogeneity may allow us to better manipulate the quantum emitters, spatially modulate exciton hybridization, and guide exciton transport. Extension of the simulations to account for non-linear effects and inhomogeneous strain might be required beyond the linear strain regime analyzed here.

Our approach to brighten and fingerprint intervalley excitons via strain engineering opens multiple possibilities for future research. First, it may be applied to identify the valley character of excitons in other systems such as moiré TMD heterostructures[57], perovskites[58,59], and 2D magnets[60,61]. Second, our technique enables optical control of spins in the Q valley that remained unexplored until now[14]. The spin/valley locked KQ excitons should exhibit a long lifetime and diffusion length, hence, may prove advantageous for spin/valley-tronics compared to the intravalley excitons[6,10,62]. We expect a pronounced strain-dependence of the transport dynamics and the lifetimes of these excitons. Third, our approach may be capable of detecting strain-dependent changes in the effective masses of excitons[24], controlling exciton-phonon interactions[27,39,44], and distinguishing different types of defects[25,26]. The application of uniaxial strain may further break the symmetry of our system, thereby changing the band topology of the corresponding excitons[63]. This enables unique prospects to manipulate valley pseudospins via large in-plane pseudo magnetic fields[64,65].

## Methods

### Sample fabrication

The devices were fabricated by dry transfer of the mechanically exfoliated TMD flakes onto a circular trench (diameter is ~5 $\mu$m) wet etched via Hydrofluoric (HF) acid in Au/Cr/SiO$_2$/Si stack. Additionally, a vent channel was designed to prevent entrapment of air inside the membrane. The strain in the membrane is induced by applying a gate voltage (typically in the range of up to ±210 V) between the TMD flake (electrically grounded) and the Si back gate of the chip. The strain in the center was characterized using laser interferometry (see Supplementary Note S4).

### Optical measurements

The devices were measured inside a cryostat (CryoVac Konti Micro) at a base temperature of 10 K. The PL measurements were carried out using the Spectrometer Kymera 193i Spectrograph, while CW lasers with $\lambda = 532$ nm (10 $\mu$W) and $\lambda = 670$ nm (6 $\mu$W) tightly focused in the center of the membrane with spot diameter ~1 $\mu$m were used to excite WS$_2$ and WSe$_2$, respectively (Supplementary Note S5). Polarization-resolved PL measurements were performed using a combination of a half-wave plate (RAC 4.2.10, B. Halle) and an analyzer (GL 10, Thorlabs) before the spectrometer to select specific polarization. The fold was confirmed using a Scanning Electron Microscopy (SEM) system Raith Pioneer II SEM/EBL at an accelerating voltage of 10 kV.

## Data availability

The data generated in this study have been deposited in the Zenodo repository under the accession code: https://zenodo.org/records/12699607.

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

## Acknowledgements

The Berlin group acknowledges the Deutsche Forschungsgemeinschaft (DFG) for financial support through the Collaborative Research Center TRR 227 Ultrafast Spin Dynamics (project B08) and the Priority Programme SPP 2244 2DMP (project BO 5142/5) as well as the Federal Ministry of Education and Research (BMBF, Projekt 05K22KE3). The Marburg group acknowledges financial support from the DFG via SFB 1083 (project B9) and the regular project 512604469. The Vienna group acknowledges financial support from The Austrian Science Fund (FWF) through the doctoral college TU-DX (DOC 142-N) and the Cluster of Excellence MECS [10.55776/COE5]. P.H.L. and S.H. acknowledge funding from the DFG under the Emmy Noether Initiative (project-ID 433878606).

## Author contributions

A.M.K., D.Y., and K.I.B. conceived the project. A.M.K., D.Y., and C.G. designed the experimental setup. A.M.K., D.Y., D.J.B., B.H., and K.B. prepared the samples. S.H. and P.H.L. developed the electrostatic straining technique. A.M.K., D.Y., and D.J.B. performed the optical measurements. R.R., J.H., and E.M. developed a theory for excitons. C.S., S.T., and F.L. developed the theory for defect states. J.N.K. performed mechanical simulations. A.M.K. and D.Y. analyzed the data. A.M.K., D.Y., and K.I.B. wrote the manuscript with input from all co-authors.

## Funding

## Competing interests

The authors declare no competing interests.
