## [Peer Review File · Nature Communications]

Exciton physics in two-dimensional semiconductors
and heterostructuresREVIEWER COMMENTS

Reviewer #1 (Remarks to the Author):

The authors present a study on the strain response of various excitonic states in WS₂ and WSe₂. They employ a setup that allows them to apply tensile strain at cryogenic temperatures by suspending the sample over a circular trench while performing optical PL measurements. Using this setup, the authors can actively track the response of several excitons to strain and “directly observe and identify previously hypothesized, but optically inaccessible intervalley KQ and ΓQ excitons”.

- One matter to address is the demonstration of the device's uniformity. Although an optical image of the monolayer WSe₂ is provided in Fig. 1c, it would be helpful if the authors could also provide a PL map of the neutral exciton in the unstrained suspended sample, both at room temperature and after pumping down.
- Another matter to address is the difficulty of distinguishing and tracking some of the excitonic peaks in the false-color map shown in Fig. 2. While Fig. 2c and Fig. 2d clearly show the peak positions, providing several line cuts at selected strain values for the device used in Fig. 2 would further clarify the spectral features and their responses to strain.
- Applying a high voltage can potentially generate heat and affect the induced strain in the sample. I wonder whether the sample temperature was monitored when applying high voltage. Additionally, was the effect of temperature considered when calculating the position-dependent strain values and simulating the strain profile in the suspended sample?

Reviewer #2 (Remarks to the Author):

Kumar et al. reported the strain fingerprints of excitonic complexes in the semiconducting two-dimensional (2D) transition metal dichalcogenides (TMDs), WSe₂ and WS₂. They demonstrated that strain could brighten the momentum-indirect excitons. In addition, excitons originating from different valleys show distinct responses to the application of strain. While binding energy is insensitive to the 0-1 % tensile strain applied in the manuscript, the observed valley-dependent shifts are attributed to the strain-dependent band structure.

I think this manuscript is well presented and the results are solid. The experimental studies on strain-engineering in 2D materials at the cryogenic temperatures has just begun in recent years. Results from Kumar et al, particularly the valley-dependent strain fingerprints, will be of broad interest to the communities of nanotechnology and condensed matter physics. As a result, I recommend the publication of this manuscript if the authors could address the following comments, most of which are about the application of strain.

1. The authors used laser interferometry measurements to determine the strain, and they chose the wavelengths of the laser to be well above the KK excitons (1s state of the A exciton). While 670 nm could be resonant to other excitonic features, such as the 2s/3s state of KK exciton and B exciton, I am wondering if the authors have considered about it.
2. The authors mentioned that the strain is a tensile strain, and Fig. 3c bottom also shows the schematic for tension. However, the schematic diagram in Fig. 1c displays the compressive strain, which could be misleading.
3. The authors are suggested to add a brief discussion on why flipping the direction of VG does not flip the sign of strain (from tension to compression).
4. The authors mentioned that the “carrier density changes with VG, estimated to be $< 1.5 \cdot 10^{12} \text{ cm}^{-2}$ in our technique, are insignificant compared to the strain-related effects.” Furthermore, they even showed more data in Fig. s2 and Fig. s4 to clarify the doping effect. However, I am not fully convinced yet for the following reasons: i) If the relative intensity of neutral and charged excitons change with VG (Fig, s4a), I suggest showing a range where carrier density is more stable. In other words, it would be better if the neutral, n-doped, and p-doped regions are not mixed in the strain plots. ii) The excitonic features change with both carrier density and strain. On the other

hand, peaks in the spectra (Fig. 2a & 2b, Fig. 3a, Fig. 4b) are quite broad. Except for KK0 and KK+, it seems that many other features are highly dependent on fitting, which could induce a large uncertainty.

5. Following the comment above, the spectrum for localized excitons (Fig. 4a) has many peaks in one frame. When strain is applied and all the peaks start to shift, it will be hard to trace all of them (P1 – P6) and make sure the shifts are accurate. Therefore, in addition to the extracted energy positions, showing the raw spectra in the supplementary could be very helpful.

6. What was the power for measuring the localized excitons?

Reviewer #3 (Remarks to the Author):

The manuscript by A. Kumar et al utilized a new nanomechanical approach developed by their previous work to study the strain effects on dark excitons in monolayer WS₂ and WSe₂. Although the data for WSe₂ is similar to their previous publication, careful analysis revealed KQ excitonic transition whose energy is almost independent from strain. They reported possible brightening of KQ exciton 1L-WSe₂ via strain-induced KK-KQ exciton hybridization, which is the first time that KQ exciton has been observed via direct optical emission, except for a recent work [Sebait, R., Rosati, R., Yun, S.J. et al. Sequential order dependent dark-exciton modulation in bi-layered TMD heterostructure. Nat Commun 14, 5548 (2023)]. For WS₂, they demonstrated distinctive strain-dependent energy shifts for KK and Γ Q excitons both theoretically and experimentally, which agree with each other. Finally, they also reported the observation of localized quantum-confined excitonic states that can be widely tuned by strain applications. Overall, the structure of this work is comprehensive. The result is convincing and could potentially be reproduced. I would like to recommend the manuscript for publication after the following minor revision.

1. I am not entirely sure about the resonance brightening of the emission from intervalley excitons. In theory, there are two pathways by which the KQ intervalley exciton can recombine: one is scattering electron to K point, and the other is scattering hole to Q point. Although the first pathway can be resonantly enhanced, the second is always non-resonant, and possibly dominant as the sample is hole doped. Clearly, in WS₂ the Γ Q exciton does not (and cannot) exhibit resonance brightening, and still acquire quite some spectral weight, so to me the experimental evidence of resonance brightening is not very strong. One may argue that the electron doping side also shows some flat feature, but it is really difficult to justify that both WSe₂ and WS₂ have exactly flat KQ energy as a function of strain, which do not agree well with first principle calculations. I would be happy with the paper without any KQ claim, unless other strong evidence/argument is added.

2. If the authors want to keep KQ, it will also be proper to move the discussion in Figure S8c panel to the main text. This would be easier for the readers to catch the possible explanation of the absence of KQ excitons in Figure 2b.

3. Can the authors please clarify with some more details how the carrier density changes being smaller than $1.5 \times 10^{12} \text{ cm}^{-2}$ was derived? Also, I agree that doping probably does not shift the peak position as much as strain, but the word "effect" is very broad and includes intensity as well, which clearly changes significantly as a function of doping. At some point, the fact that KK+ is primarily brightened by p-doping should be mentioned in the manuscript.

Reviewer #4 (Remarks to the Author):

REVIEWER COMMENTS

We thank all reviewers for their help in clarifying and improving the message of the manuscript. We appreciate the feedback and hope that our response fully answers all questions. For the revised manuscript, we performed additional measurements, numerical simulations, and theory calculations. We believe that this improved our understanding of heating, doping, and strain effects in our system. In the following, we provide a point-by-point response (black) to the reviewer's questions (blue) and corresponding modifications to the manuscript (red).

Reviewer #1 (Remarks to the Author):

The authors present a study on the strain response of various excitonic states in WS₂ and WSe₂. They employ a setup that allows them to apply tensile strain at cryogenic temperatures by suspending the sample over a circular trench while performing optical PL measurements. Using this setup, the authors can actively track the response of several excitons to strain and “directly observe and identify previously hypothesized, but optically inaccessible intervalley KQ and Γ Q excitons”.

We thank the reviewer for devoting their time and careful assessment of our work.

- One matter to address is the demonstration of the device's uniformity. Although an optical image of the monolayer WSe₂ is provided in Fig. 1c, it would be helpful if the authors could also provide a PL map of the neutral exciton in the unstrained suspended sample, both at room temperature and after pumping down.

We followed this suggestion and measured a spatial PL map in a suspended monolayer WSe₂ inside the cryostat before (Fig. R1a) and after (Fig. R1b) pump down (with a step size of 0.5 μ m at a laser power of 7 μ W). To assess the strain effects, we plot the position-dependent energy shift of the neutral exciton peak with respect to the membrane center. The standard deviation of the peak position in a circular area with a diameter of 3 μ m around the center of the membrane (dashed line) does not exceed 3 meV. This suggests that the pre-strain non-uniformity (quantified by the standard deviation) is below 0.05%, which is close to the sensitivity of our method. The shift of the exciton peak reaches \sim 8 meV near the edges of the membrane (solid line). This is likely influenced by a change in the dielectric environment (different dielectric constants of air and gold). We note that there is no detectable change in pre-strain upon pump-down. Hence, we assume that the pre-strain is nearly uniform and serves as a constant offset to the applied strain.

Fig. R1. Spatial map of the energy shift of the neutral exciton PL peak for a representative monolayer WSe₂ device, shown in ambient conditions (a) and after pump-down (b). The solid circle marks the trench edge, while the dashed circle indicates the region of homogeneous strain. The higher exciton emission energy in the suspended region compared to the supported region is attributed to differences in the dielectric environment.

We highlight one aspect of our device's design that helped achieving homogenous strain. In general, air trapped under the membrane expands after pump down, changing the uniformity of the strain and potentially damaging the device, as noted by the reviewer. To address this challenge, we implemented a "vent channel" in our device connecting the area under the membrane with the atmosphere (Fig. R2, also Methods in the main manuscript). This vent channel allows pressure equilibration during pump down.

Fig. R2. Top: optical image of a typical sample. Bottom: side view schematic of the venting channel, which prevents air entrapment under the suspended sample. The channel, stopping short of the sample, increases strain uniformity and connects to the suspended region after wet etching due to under-etching effect.

We have included Fig. R1 in the supplementary information as Fig. S6 as well as a description of device uniformity in supplementary section S5:

Device uniformity is influenced by i) pre-strain, and ii) spatial strain-gradient under high gate voltages.

To assess device uniformity and pre-strain consistency, we measured spatial PL maps at room temperature (Fig. S6a). Our analysis of the spatial variation in the excitonic energy confirms a small strain inhomogeneity of 0.03% in the pristine device. The same observation was recorded when the device was pumped down (Fig. S6b), confirming a uniformity in the pre-strain. Additionally, we insured device stability against any pocket of air trapped air inside the cavity by implementing a vent channel for pressure equilibration (see Methods in the main text).

The strain inhomogeneity increases under the application of large gate voltages. Using a combination of COMSOL simulation and spatial mapping of neutral exciton peak, we find that within 1 μm from the center of the membrane, the strain decreases to $0.9 \cdot \epsilon_{center}$ (Fig. S6d,e). Note, that the degree of homogeneity can be controlled by controlling the size of the circular trench. To optimize our experiments, we ensure that the laser spot (diameter $\sim 1 \mu\text{m}$) is tightly-focused in the center of the membrane.

- Another matter to address is the difficulty of distinguishing and tracking some of the excitonic peaks in the false-color map shown in Fig. 2. While Fig. 2c and Fig. 2d clearly show the peak positions, providing several line cuts at selected strain values for the device used in Fig. 2 would further clarify the spectral features and their responses to strain.

We agree. We have added selected line cuts from Fig. 2a and Fig. 2c along with the fits corresponding to individual peaks to Fig. R3 (also, in Fig. S2 in supplementary information).

Fig. R3. Selected PL line cuts from WSe₂ and WS₂ corresponding to the data in Fig. 2 in the main text. Dashed lines highlight individual peaks obtained from fits, while cumulative fits are semi-transparent solid lines. Note that some low energy KK excitons are doping-dependent, and are prominent only in low-strain regime. This serves as an additional source of uncertainty in the fitting procedure. In our analysis, we considered features that persisted for the entire range of V_G (Fig. 2 in the main text).

Additionally, in Supplementary Note S5, we provide a description of the fitting procedure utilized for precise estimation of the exciton energy positions:

We use the second derivative of the PL intensity, $d^2\text{PL}/dE^2$, to obtain an initial guess for the PL peak positions from $E_{\text{center}} = \max(d^2\text{PL}/dE^2)$ and amplitude $A = \text{PL}(E_{\text{center}})$. We then fit the raw PL(E) spectra with several Gaussians using the initial guesses obtained above. We perform batch fitting to minimize selective biases during the fitting procedure. Finally, we estimate uncertainties (shadows in Fig. 2c,d) by varying the constraints/parameters of the fit (e.g. number of peaks used in fitting, peak-width, ascending and descending order of strains during fitting etc).

- Applying a high voltage can potentially generate heat and affect the induced strain in the sample. I wonder whether the sample temperature was monitored when applying high voltage. Additionally, was the effect of temperature considered when calculating the position-dependent strain values and simulating the strain profile in the suspended sample?

We agree that the heating of our device may potentially influence the strain via thermo-mechanical coupling. To address this concern, we estimate heating due to the application of high gate voltage. Power generated due to an applied voltage V_G can be estimated using Joule's law, $P = VI$, where I is the leakage current and V is the voltage. In our devices, we applied up to V_G = 200 V, at which point the current (which we monitor throughout the experiments) remains well below 10 nA. This means that the maximum dissipated power remains below 2 μW, nearly an order of magnitude less than the laser power used for PL measurements, up to 10 μW. Moreover, unlike the laser power that is dissipated predominantly within the membrane, the Joule power is dissipated inside the entire area of the chip (between the bottom conductive Silicon and the top conductive gold electrodes, across the SiO₂ gate dielectric). The membrane covers only ~0.1% of this area. We therefore believe that heating due to spurious leakage currents is negligible compared to laser-related heating.

Next, we estimate the amount of laser-induced heating, the largest source of spurious heating in our system. These estimates also address the reviewer’s question about position-dependent strain. In contrast to Joule heating, the laser absorption directly heats the membrane within a confinement region of $\sim 1 \mu\text{m}$. We performed numerical simulations to quantify the laser-related temperature increase and strain change due to the thermal expansion of WSe_2 . Assuming a positive thermal expansion coefficient of $7.3 \times 10^{-6} \text{ K}^{-1}$ [Y. Zhong, et al. *Sci. Adv* 8, eabo3783 (2022); X. Hu, et al. *Phys. Rev. Lett.* 120, 055902 (2018)], we obtain a decrease of strain of 0.06% in the center of the membrane ($T = 5 \text{ K}$). The temperature and strain relax away from the membrane center (Fig. R4). Importantly, this strain change can be treated as a static offset to the built-in strain, and is much smaller than the maximum strain produced by gating both at $T = 5 \text{ K}$ and $T = 293 \text{ K}$.

Fig. R4. (a),(b) Laser heating-induced strain change profile in a suspended sample exposed to $10 \mu\text{W}$ of laser power at $T = 293 \text{ K}$ and 5 K , respectively, simulated in COMSOL Multiphysics. The heating-induced strain change is maximum in the center of the membrane and remains nearly uniform within $\pm 1 \mu\text{m}$ region. The heating effects are more pronounced at 5 K . (c) Spatial line cut of strain profile through the membrane with (blue) and without (black) the consideration of heating effects. While the net strain decreases, the heating effect remains nearly constant for all applied gate voltages. Gate-dependent excitonic peak shifts, the main observable analyzed in the manuscript, are therefore unaffected by the heating.

We have added Fig. R4b to Supplementary figure Fig. S6 and added a Note S5 about possible artefacts due to heating effects:

The strain response in our devices may be affected by heating effects leading to undesirable thermal expansion or contraction. Heating can arise from either the application of large gate voltages or laser absorption by the membrane. The former effect is negligible since the current generated in our capacitor-like device stays below 10 nA even for a gate voltage exceeding 150 V that dissipates over the entire area of the conductive pad. In contrast, laser absorption in the center of the membrane locally increases the sample temperature. This, in turn, relaxes the net strain in the membrane due to a positive thermal expansion coefficient of the TMDs²⁸. For a precise estimation, we simulate the laser heating-induced strain profile in a monolayer WSe_2 device at $T = 5 \text{ K}$ (Fig. S6c) and find the strain variation to be on the order of the pre-strain in our device.

In our study, we approximate the laser heating effect as a static strain component comparable to the pre-strain. Although the cavity interference effects and the band structure modulation may change the absorbed laser power, the related strain magnitude is nearly constant and remains.

Reviewer #2 (Remarks to the Author):

Kumar et al. reported the strain fingerprints of excitonic complexes in the semiconducting two-dimensional (2D) transition metal dichalcogenides (TMDs), WSe_2 and WS_2 . They demonstrated that strain could brighten the momentum-indirect excitons. In addition, excitons originating from different

valleys show distinct responses to the application of strain. While binding energy is insensitive to the 0-1 % tensile strain applied in the manuscript, the observed valley-dependent shifts are attributed to the strain-dependent band structure.

I think this manuscript is well presented and the results are solid. The experimental studies on strain-engineering in 2D materials at the cryogenic temperatures has just begun in recent years. Results from Kumar et al, particularly the valley-dependent strain fingerprints, will be of broad interest to the communities of nanotechnology and condensed matter physics. As a result, I recommend the publication of this manuscript if the authors could address the following comments, most of which are about the application of strain.

We thank the reviewer for their careful analysis of our work.

1. The authors used laser interferometry measurements to determine the strain, and they chose the wavelengths of the laser to be well above the KK excitons (1s state of the A exciton). While 670 nm could be resonant to other excitonic features, such as the 2s/3s state of KK exciton and B exciton, I am wondering if the authors have considered about it.

We have considered this effect. The cavity interference measurements rely on optical constants of WSe₂ such as absorption/reflection coefficients that become strain-dependent due to a continuous modulation in the excitonic band structure. To quantify the strain-dependent changes in the membrane's reflectance and absorbance, we first estimate strain-induced changes in the real and imaginary parts of the dielectric function of WSe₂ and then use the transfer matrix method to extract the reflectance and absorbance. Assuming that the oscillator strength of A and B excitons remains constant within the strain range considered here, the effect of strain (ε) on the dielectric function (n) is limited to the shift in energy:

$$n(E_{laser}, \varepsilon) = n(E_{laser} - \Omega \varepsilon, 0),$$

where Ω is the strain gauge factor and E_{laser} is the laser photon energy.

In Fig. R5, we plot the reflectance and absorbance obtained from $n(E_{laser}, \varepsilon)$ for suspended WS₂ (blue line) and WSe₂ (red line) in our geometry, corresponding to the laser energy of 2.33 eV (532 nm) and 1.85 eV (670 nm), respectively [C. Hsu, et al. Adv. Optical Mater. 7, 1900239 (2019)]. We find that both reflectance and absorbance are on the order of 2%, and are well approximated by a linear function in strain.

We incorporate these effects in our analysis of the interferometric measurements. Specifically, in the Eq. 7 of Section S4, we model the reflected light intensity ΔI as:

$$\Delta I_{laser} \sim A(V_G) \sin^2 \left(4\pi \cdot \frac{d(V_G)}{\lambda} + \phi \right)$$

where, the term $A(V_G) = A_0 + A_1 V_G^2$ describes the voltage-dependent changes in the optical constants. Indeed. Our fits show that the second term in $A(V_G)$ is indeed a small correction: $\max(A_1 V_G^2)/A_0 \sim 15\%$

We have updated the text in section S4 as follows:

Since the laser energy used in our experiments is ~ 100 and ~ 220 meV away from the excitonic resonance in WSe_2 and WS_2 , respectively, a strain-induced modulation in the excitonic band structure may slightly change the optical constants. Therefore, we assume the pre-factor in Eq. 7 to be V_G dependent, and approximate this as $A(V_G) = A_0 + A_1 V_G^2$ via a Taylor expansion. Note that there is no linear term in the expansion since the force acting on the flake – and hence all optical constants – are even functions of V_G .

Fig. R5. Reflectance **(a)** and absorbance **(b)** of WS_2 (blue line) and WSe_2 (red line). Within the applied strain range, the reflectance and absorbance of suspended flakes remain below 3% and are well approximated by a linear dependence (dashed lines).

2. The authors mentioned that the strain is a tensile strain, and Fig. 3c bottom also shows the schematic for tension. However, the schematic diagram in Fig. 1c displays the compressive strain, which could be misleading.

We interpret the reviewer's comment as compressive strain being associated with the apparent downward deflection of the membrane. We acknowledge any confusion caused by the cartoon in Fig. 1c, and, in the following, elucidate the nature of strain in our device.

Our device geometry is essentially a plate capacitor where an applied gate voltage V_G induces planes of positive and negative charges across the dielectric. The charge distribution results in an effective electrostatic pressure on the suspended membrane, and hence, the membrane is pulled downward under an attractive electrostatic force. Consequently, the membrane deformation causes a biaxial tensile strain in the center of the membrane. Nonetheless, we admit that Fig. 1c is an exaggerated cartoon representation of the straining mechanism. To improve clarity, we have added arrows to highlight the tensile strain in the center of the membrane.

3. The authors are suggested to add a brief discussion on why flipping the direction of V_G does not flip the sign of strain (from tension to compression).

We appreciate the reviewer for highlighting the importance of clarifying the nature of strain in our device. In our plate-capacitor-like device geometry, a gate voltage V_G applied across the dielectric between the gate and the sample induces opposite charge densities on both "plates" (the gate and the membrane). Critically, the electrostatic interaction between the two oppositely charged plates is attractive and is independent of the sign of V_G . Therefore, the suspended membrane is pulled downward for both polarities of V_G and the strain remains tensile. While the sign of the gate voltage controls whether electrons or the holes are injected into the TMD membrane (leading to an unwanted artefact in the context of our work), the tensile nature of strain in our device is confirmed via excitonic redshift for both polarities of V_G (Fig. S2). We have modified the text in the manuscript to clarify the effect of the sign of V_G on strain:

The induced strain is tensile and symmetric vs. polarity of V_G due to the attractive nature of the electrostatic force (Fig. S2).

4. The authors mentioned that the “carrier density changes with V_G , estimated to be $< 1.5 \cdot 10^{12} \text{ cm}^{-2}$ in our technique, are insignificant compared to the strain-related effects.”

We realize the inaccuracy in this statement and have made the language more precise. The sentence now reads:

...carrier density-related changes in exciton emission energy are an order of amplitude smaller compared to strain ($\Delta E(n) < 10 \text{ meV}$ for $n < 1.5 \cdot 10^{12} \text{ cm}^{-2}$, see Fig. S4d and Note S6)...

Furthermore, they even showed more data in Fig. s2 and Fig. s4 to clarify the doping effect. However, I am not fully convinced yet for the following reasons: i) If the relative intensity of neutral and charged excitons change with V_G (Fig. s4a), I suggest showing a range where carrier density is more stable. In other words, it would be better if the neutral, n-doped, and p-doped regions are not mixed in the strain plots. ii) The excitonic features change with both carrier density and strain. On the other hand, peaks in the spectra (Fig. 2a & 2b, Fig. 3a, Fig. s4b) are quite broad. Except for KK_0 and KK_+ , it seems that many other features are highly dependent on fitting, which could induce a large uncertainty.

While we agree that coupling between strain and changes in carrier density limits the flexibility of our approach, we believe that the effects of strain and carrier density can be reliably disentangled. Although the excitonic features change with both doping and strain, our work focuses on the energy shift of the excitonic peaks. Motivated by the reviewer’s comment, we analyzed three different approaches to disentangle the two contributions: i) by comparing samples with different doping levels, ii) by analyzing V_G dependence of excitonic peaks, and iii) by analyzing the magnitude of changes.

Comparing samples with different doping levels. Following the reviewers’ comment, we measured and compared 2 new samples with different doping levels in addition to the sample analyzed in the main text (Fig. R6).

Fig. R6. a-c): Comparison of strain dependence in three different samples (#1,#2, #3) with intrinsic doping of $n_0 = 0.2 \cdot 10^{12}$; $0.3 \cdot 10^{12}$ and $0.4 \cdot 10^{12} \text{ cm}^{-2}$, respectively. Sample #1 is the one analyzed in the main text; samples #2 and #3 are new.

These three samples have different intrinsic doping levels and, correspondingly, the shift in excitonic energy vs. V_G should be different. At the same, the strain depends only on V_G and the built-in strain. Importantly, while the charge neutrality point may vary across devices, all excitonic features shift at the same rate in all 3 devices. This behavior is consistent with our assertion that V_G -dependent changes in peak positions at high V_G are predominantly due to strain and strain fingerprinting of excitons is a universal feature.

Analyzing V_G dependence. Carrier-density-related effects are approximately linear with V_G (and slightly super-linear with V_G for large membrane displacement):

$$n = C_G V_G + n_0 \quad (1)$$

Conversely, strain is a non-linear function of V_G [Nat Commun 13, 7691 (2022)]:

$$\varepsilon \approx \begin{cases} \alpha V_G^4 + \varepsilon_0, & \alpha V_G^4 \ll \varepsilon_0 \\ \beta V_G^{4/3}, & \alpha V_G^4 \gg \varepsilon_0 \end{cases} \quad (2)$$

A significantly larger sensitivity of strain to V_G compared to doping allows us to separate the two effects. To visualize this difference, PL spectra are plotted below vs. strain and V_G (Fig. R7a and R7b, respectively). We expect carrier-density-related shifts to be linear in PL vs. V_G graph (see Eq.1; also Eq. 4 in response to reviewer 3). However, we see that most of the features shift linearly only in PL vs. strain (that is derived through an independent interferometric measurement). Moreover, the strain gauge factors for different states from the same valley (e.g. different K valley excitons) show a variation below 10% which has contributions from doping effects, strain-dependent changes in phonon energy and effective masses, and experimental uncertainties. This again suggests that strain-related effects dominate over doping-related effects.

Fig. R7. PL spectra of the same sample plotted vs. strain **(a)** and vs. V_G **(b)**.

Analyzing the magnitude of changes. The dominant effect of a change in carrier density on excitonic peak position arises from the shift of the Fermi energy level that modifies exciton-trion energy separation. Hence, we split the analysis of excitonic peaks into two regimes of V_G : i) $|V_G| < 30$ V, and ii) $|V_G| > 30$ V. In the first regime, doping effects dominate strain-related changes: a relative energy separation of the exciton-trion peak increases by ~ 3 meV, which corresponds to a carrier density change of $\sim 0.4 \cdot 10^{12} \text{ cm}^{-2}$ (see the response to question 3 of reviewer 3). In the second regime, strain-related energy shifts dominate. We find that a relative energy shift in exciton-trion separation due to the carrier density changes ($< 1.5 \cdot 10^{12} \text{ cm}^{-2}$) does not exceed 10 meV, an order of magnitude below than the strain-induced energy shift, > 100 meV. We note that decoupling the doping and strain effects for other excitonic features such as intensity and linewidth may be complex, and is not within the scope of the present work.

We also agree with the comment that “*it would be better if the neutral, n-doped, and p-doped regions are not mixed in the strain plots.*”. Since, in the interest of transparency, we prefer to show the full measured data, we added explicit labeling marking these regions to the PL vs. V_G in the manuscript (see, e.g., Fig. S4a).

Finally, the reviewer expressed their concern regarding the broad excitonic linewidth, which might result in a large energy uncertainty via the fitting procedure. We agree with the reviewer that, indeed, the excitonic peaks in our devices are broader compared to the state-of-the-art hBN-encapsulated devices. These linewidths further increase due to effects such as strain-related exciton hybridization and spatial strain inhomogeneity. To ensure that these uncertainties are thoroughly considered in our analysis, we have added the following note to the section S5:

We use the second derivative of the PL intensity, $d^2\text{PL}/dE^2$, to obtain an initial guess for the PL peak positions from $E_{\text{center}} = \max(d^2\text{PL}/dE^2)$ and amplitude $A = \text{PL}(E_{\text{center}})$. We then fit the raw PL(E) spectra with several Gaussians using the initial guesses obtained above. We perform batch fitting to minimize selective biases during the fitting procedure. Finally, we estimate uncertainties (shadows in Fig. 2c,d) by varying the constraints/parameters of the fit (e.g. number of peaks used in fitting, peak-width, ascending and descending order of strains during fitting etc).

5. Following the comment above, the spectrum for localized excitons (Fig. 4a) has many peaks in one frame. When strain is applied and all the peaks start to shift, it will be hard to trace all of them (P1 – P6) and make sure the shifts are accurate. Therefore, in addition to the extracted energy positions, showing the raw spectra in the supplementary could be very helpful.

As suggested, we have added the raw PL spectra at selected strain values in Fig. S10 and also in Fig. R8, below. Indeed, as the states begin to shift under strain, a precise extraction of peak energy becomes challenging due to state mixing, leading to a large uncertainty in the energy shift. Consequently, we now add an uncertainty bar to the energy shift of quantum-confined excitons:

Crucially, the energy of emitters P1–P6 can be tuned by up to 60 meV under the application of strain.

Fig. R8. a) PL map from the device that shows Quantum-confined excitons. b) PL spectra at selected strain values. Several sharp features are seen. We selectively highlight 6 features, color-coded with respect to the data in Fig. 4a in the main text. Overall, we note complex excitonic features that make it challenging to identify the energy shift of localized excitons.

6. What was the power for measuring the localized excitons?

The power employed for these measurements was $6\ \mu\text{W}$. This specific power level was chosen to minimize the heating of the membrane (see the response to the last question of reviewer 1 for the analysis of heating-related artefacts) while maintaining a sufficient signal-to-noise ratio. To clarify the effect of laser power on the strain response of our sample we have added a Supplementary Note S5:

To assess device uniformity and pre-strain consistency, we measured spatial PL maps at room temperature (Fig. S6a). Our analysis of the spatial variation in the excitonic energy confirms a small strain inhomogeneity of 0.03% in the pristine device. The same observation was recorded when the

device was pumped down (Fig. S6b), confirming a uniformity in the pre-strain. Additionally, we insured device stability against any pocket of air trapped air inside the cavity by implementing a vent channel for pressure equilibration (see Methods in the main text).

Reviewer #3 (Remarks to the Author):

The manuscript by A. Kumar et al utilized a new nanomechanical approach developed by their previous work to study the strain effects on dark excitons in monolayer WS₂ and WSe₂. Although the data for WSe₂ is similar to their previous publication, careful analysis revealed KQ excitonic transition whose energy is almost independent from strain. They reported possible brightening of KQ exciton 1L-WSe₂ via strain-induced KK-KQ exciton hybridization, which is the first time that KQ exciton has been observed via direct optical emission, except for a recent work [Sebait, R., Rosati, R., Yun, S.J. et al. Sequential order dependent dark-exciton modulation in bi-layered TMD heterostructure. Nat Commun 14, 5548 (2023)]. For WS₂, they demonstrated distinctive strain-dependent energy shifts for KK and Γ Q excitons both theoretically and experimentally, which agree with each other. Finally, they also reported the observation of localized quantum-confined excitonic states that can be widely tuned by strain applications. Overall, the structure of this work is comprehensive. The result is convincing and could potentially be reproduced. I would like to recommend the manuscript for publication after the following minor revision.

We thank the reviewer for their thorough assessment of our work and suggestion of a valuable paper, which we have now cited.

1. I am not entirely sure about the resonance brightening of the emission from intervalley excitons. In theory, there are two pathways by which the KQ intervalley exciton can recombine: one is scattering electron to K point, and the other is scattering hole to Q point. Although the first pathway can be resonantly enhanced, the second is always non-resonant, and possibly dominant as the sample is hole doped. Clearly, in WS₂ the Γ Q exciton does not (and cannot) exhibit resonance brightening, and still acquire quite some spectral weight, so to me the experimental evidence of resonance brightening is not very strong. One may argue that the electron doping side also shows some flat feature, but it is really difficult to justify that both WSe₂ and WS₂ have exactly flat KQ energy as a function of strain, which do not agree well with first principle calculations. I would be happy with the paper without any KQ claim, unless other strong evidence/argument is added.

We agree with this criticism and add new evidence and arguments. We start by summarizing our previous observations, followed by additional experiments, simulations and arguments that we developed during the review process.

The main argument of our work is that all intervalley excitons (KQ, Γ Q, Γ K) have specific strain gauge factors different than the intravalley (KK) excitons. We did observe excitonic features with gauge factors matching those of KK, Γ Q in monolayers and features matching the gauge factor of KQ in a bilayer WSe₂. These features are likely visible due to indirect recombination (“non-resonant pathway”) enabled by either phonon-assisted or a scattering mechanism breaking the translational symmetry, e.g. impurity scattering. However, features with the gauge factor of the KQ excitons do not appear in monolayers. Instead, we see a new excitonic feature at 1.7 eV with zero strain gauge factor. This gauge factor is not expected for any free exciton. We hypothesize that the new feature is *a KQ exciton resonantly brightened via coupling to a KK exciton*. In our model (based on first-principle calculations), the apparent zero-gauge factor is explained by the resonant enhancement of that exciton. That emission is only observed from a region in the membrane *where the local strain is $\sim 0.35\%$, the value at which KK and KQ excitons are in energetic resonance*. The slight strain inhomogeneity in our devices means that when the entire device is strained, the local region where 0.35% of strain is reached first appears in the center of the device and then moves outwards as the membrane is stretched further. Critically, *the spectral position of the feature in this model always remains the same*. Our model correctly predicts the following aspects of the feature at 1.7 eV:

- **Energy position.** The peak at 1.7 eV only appears when the strain level is enough to bring KQ and KK excitons into energetic resonance.
- **Strain range.** Since KQ is bright only when it is hybridized with KK exciton, the emission energy of the exciton is effectively strain-independent and only appears when the hybridization conditions are met.
- **Intensity.** The intensity of the 1.7 eV peak decreases monotonically as V_G is increased. This is consistent with our expectation of the local resonance point being shifted away from membrane's center when V_G is increased (Fig. S7 in the Supplementary Info).
- **Carrier density independence.** The peak appears for both polarities of V_G , corresponding to hole and electron (Fig. R9) doping. This agrees with our model in which KK/KQ resonance is triggered by strain and is not dependent on the carrier density. Conversely, some non-resonant brightening mechanisms can be carrier density-dependent.

Fig. R9. KQ exciton in the n-doping regime. PL spectra of WSe₂ monolayer from 110 V to 160 V. Similar to the p-doping regime (see Fig. 3a of the manuscript), a new peak at 1.7 eV appears around 0.3% strain in the n-doping regime, suggesting the brightening mechanism is independent of doping.

In addition, we note that the accuracy of the presented first-principle calculation for KQ excitons (on which our model relies) is validated by their observation in bilayer WSe₂ (see Fig. 3f in the main manuscript), with gauge factors consistent with the model, incorporating a non-resonant brightening mechanism.

To further test our model, we carried out a new experiment testing its key prediction: a non-uniform spatial distribution of the KQ emission. As mentioned earlier, the resonant emission is only expected from regions where the local strain is $\epsilon \approx 0.35\%$ and thus KK/KQ resonant hybridization conditions are locally met. This contrasts with non-resonant emission, where the intensity is expected to be relatively homogenous.

We measured the spatial profile of KQ emission in monolayer WSe₂ (Fig. R10). We normalize the KQ intensity with respect to the KK exciton to rule out effects related to cavity interference. The strain at the center of the membrane is 0.65%. We observe a clear increase in relative KQ emission away from the membrane center. This is exactly what is expected in our model – the strain in the center of the membrane is too high for the hybridization to be effective. Conversely, the strain becomes spatially inhomogeneous and drops off towards the membrane's edges. Eventually, the strain approaches 0.35% close to the membrane's edges, and the hybridization condition is met, resulting in emission from that region (see inset of Fig. R10). We note that while an enhancement of the non-resonant emission of the KQ exciton away from the membrane center (i.e. decreasing local strain) cannot be

ruled out completely, our observations completely follow the expected dependence of spatially varying KK-KQ hybridization.

Fig. R10. Spatial dependence of KQ emission. Relative KQ intensity in 1L-WSe₂ with respect to the X_{KK}^0 across the membrane position when the strain in the center was 0.65%. The red shadow is the error bar, while gray shadows correspond to the region where the edge effect becomes dominant and extracting relative intensities is challenging. The inset shows a cartoon representation of KQ emission via resonant brightening which is away from the center.

The arguments presented above rely on one key assumption: the efficiency of resonant brightening of KQ excitons hybridized with KK excitons is higher than that of non-resonant recombination of KQ excitons. To verify this, we performed additional calculations to disentangle the resonant and non-resonant contributions. Figure R11 shows the intensity of resonant (due to coupling to KK) and non-resonant (phonon-activated) emission from KQ. We find that at $T = 20$ K, the non-resonant emission of KQ with respect to KK decreases by up to 6 orders of magnitude within 0.2% strain (dark red line in Fig. R11). This is due to thermal occupation, which reduces the occupation of the KQ state as it moves up in energy with the strain. Conversely, resonant emission increases sharply near 0.35% and reaches a maximum before showing a monotonic decay with increasing strain (bright red line). Overall, these observations support our argument that our observation of the KQ exciton is dominated by the resonant emission mechanism. We have added Fig. R10 and R11 into the Supplementary figure Fig. S7.

Fig. R11. KQ exciton emission via resonant and non-resonant vs. resonant brightening. The solid lines show the calculated emission intensity of the KQ exciton, considering only non-resonant (dark red) and resonant (bright red) effects. The measured intensity of the KQ exciton is represented by points, with the red shading indicating the uncertainty.

To summarize the discussion above, while we fully share reviewers' concerns regarding resonant- vs. non-resonant brightening mechanisms, we believe that the data we presented reach the level of certainty necessary to claim the observation of a KQ state. At the same time, more effort is needed to understand, especially, the non-resonant emission mechanisms. To reflect this view, we made the following changes to the manuscript:

- i) We refer to the resonant brightening scenario as one of the possible brightening mechanisms:
In addition to the resonant brightening, a KQ exciton emission may originate from non-resonant mechanism, e.g. phonon assisted emission or scattering on a defect, which is not included in our model [41]. However, the distinctive traits of a KQ exciton such as the unique strain response of the emission energy and intensity are accurately captured by our model, and cannot be explained by a non-resonant mechanism (Fig. S7).
- ii) We present the additional experimental and theoretical results mentioned above in the Supplementary Info, Fig. S7c,d.
- iii) We removed the claim of KQ excitons in WS_2 (see answer to the next question)

2. If the authors want to keep KQ, it will also be proper to move the discussion in Figure S8c panel to the main text. This would be easier for the readers to catch the possible explanation of the absence of KQ excitons in Figure 2b.

Considering all additional experimental and theoretical evidence of KK/KQ hybridization in WSe_2 , we believe to have a high degree of confidence in the KQ observation in WSe_2 . However, we cannot reach the same level of confidence in another material that we study, WS_2 . In that material, our investigations were challenged by i) a bright Γ Q exciton persisting beyond $\varepsilon \sim 0.3\%$, required for KK-KQ hybridization, ii) spatial strain inhomogeneity causing a tail on the higher energy side, iii) doping effects, and iv) energetic vicinity to the Γ K exciton that is expected to appear in the strain regime, $>1\%$.

While we consistently observed a strain-independent feature with weak intensity, we could not resolve the brightness, spatial distribution, and polarization of this peak and compare with the KK excitons.

Therefore, we followed the reviewer's suggestion and have removed the claim of KQ excitons in WS₂:

We also observe signatures of a similar peak in 1L-WS₂, however, in that material, an energetic proximity of KQ, ΓQ, and ΓK excitons makes the hybridization mechanism more intricate (Fig. S8).

We label the strain independent and linear in power feature in WS₂ as Y-peak and suggest that future investigations are required to resolve the complex excitonic picture in this strain regime.

3. Can the authors please clarify with some more details how the carrier density changes being smaller than 1.5e12 cm⁻² was derived?

The change in the carrier density ($n_{e,h}$) induced by the gate voltage in the center of the membrane is estimated using a plate capacitor model:

$$n_{e,h} = \frac{(V_G - V_0)\epsilon_0}{e} \left(\frac{\epsilon_{SiO_2}}{d_{SiO_2} + \epsilon_{SiO_2}(d_{Au} - d(V_G))} \right) \quad (3)$$

Here ϵ_0 and $\epsilon_{SiO_2} = 3.6$ are the vacuum permittivity and the dielectric constant of SiO₂, respectively. $V_0 = 55$ V is the gate voltage at which the sample is charge neutral. $d_{SiO_2} = 900$ nm is the SiO₂ thickness, $d_{Au} = 600$ nm is the distance between the gold surface and SiO₂, $d(V_G)$ is the flake deflection (obtained from interferometry) and e is the elemental charge. From this method, we obtain $n_{e,h} \sim 0.8 \cdot 10^{12}$ cm⁻² at $V_G = -150$ V.

The accuracy of this method is significantly challenged by artefacts due to Schottky barriers, defects, photo-doping, etc., resulting into a nonlinear dependence of $\Delta n_{e,h}$ on V_G . However, the experimental features of excitons and trions (intensity, energy separation, linewidth, etc.) carry an accurate measure of Fermi energy level shift, regardless of the artefacts present in the device. In particular, the exciton-trion energy separation (ΔE_{XT}) relates to the Fermi energy shift, as $\Delta E_{XT} \approx E_F$ [PRL 125, 267401 (2020)]. The carrier density, in turn, relates to the Fermi energy as

$$n_{e,h} = \frac{E_F m_{e,h}}{\pi \hbar^2} \quad (4)$$

Here $m_{e(h)} = 0.36$ (0.40) m_0 is the effective mass of an electron (hole) [2D Mater. 2 022001 (2015)]. Using Eq. 4 we estimate the carrier density to be $n_h \approx 1 \cdot 10^{12}$ cm⁻² at 1% strain (Fig. S4d in the Supplementary Information and Note S6). We find a variation in the induced carrier density by up to 30% across different approaches (e.g. trion/Fermi-polaron approximation in Eq. 4).

We note that for strain values > 0.8 % ($V_G < -140$ V, see Fig. S2a), state mixing, broader peaks, and a decreasing neutral exciton intensity significantly increase the uncertainty in ΔE_{XT} . To verify the accuracy of our method, in Fig. R13, we compare the carrier density extraction using both the methods

discussed above. Overall, we find that both methods result in carrier density changes on the same order of magnitude.

Fig. R13. Comparing different models to estimate carrier density. The solid line is the carrier density estimated via the capacitor model. Circles correspond to the method (eq. 4) applied in our work.

To make this point clear to the reader, we have updated Fig. S4d to reflect an improved accuracy of carrier density.

Also, I agree that doping probably does not shift the peak position as much as strain, but the word “effect” is very broad and includes intensity as well, which clearly changes significantly as a function of doping. At some point, the fact that KK^+ is primarily brightened by p-doping should be mentioned in the manuscript.

To address this valid concern, we modified the text as follows:

...carrier density-related changes in exciton emission energy are an order of amplitude smaller compared to strain ($\Delta E(n) < 10 \text{ meV}$ for $n < 1.5 \cdot 10^{12} \text{ cm}^{-2}$, see Fig. S4d and Note S6)...

We note that carrier density changes influence the intensity and linewidth of the excitonic features (Fig. S4), which is outside the scope of our study.

Reviewer #4 (Remarks to the Author):

We thank reviewer 4 for devoting time to help with the review of our manuscript.

REVIEWERS' COMMENTS

Reviewer #1 (Remarks to the Author):

My comments have been thoroughly addressed. The authors have carefully revised and improved the manuscript, and I now recommend it for publication.

Reviewer #2 (Remarks to the Author):

The authors have provided convincing and detailed answers to all of my questions. I support the publication of this manuscript on Nature Communications.

Reviewer #3 (Remarks to the Author):

The revision by Kumar et al completely addressed my comment. Therefore I would like to recommend it for publication.

Reviewer #4 (Remarks to the Author):

The authors address my questions well and I would recommend for publication.

RESPONSE TO REVIEWERS' COMMENTS

Reviewer #1 (Remarks to the Author):

My comments have been thoroughly addressed. The authors have carefully revised and improved the manuscript, and I now recommend it for publication.

We thank the reviewer for their valuable feedback on the manuscript and for their support regarding the publication of our work.

Reviewer #2 (Remarks to the Author):

The authors have provided convincing and detailed answers to all of my questions. I support the publication of this manuscript on Nature Communications.

We thank the reviewer for their critical feedback and for their positive recommendation for the publication of our work.

Reviewer #3 (Remarks to the Author):

The revision by Kumar et al completely addressed my comment. Therefore I would like to recommend it for publication.

We thank the reviewer for their valuable contributions in reviewing our manuscript and for their recommendation to publish this manuscript.

Reviewer #4 (Remarks to the Author):

The authors address my questions well and I would recommend for publication.

We thank the reviewer for devoting their time in reviewing our work and positive appreciation.